# The Effect of Home-Based Robotic Rehabilitation on Individuals with Disabilities in Community Settings: A Pilot Study

**DOI:** 10.3390/healthcare13010078

**Published:** 2025-01-04

**Authors:** Joonhwan Lee, Eunyoung Lee, Seokjoon Hong, Sunyi Shin, Byungju Ryu

**Affiliations:** 1Department of Physical Medicine and Rehabilitation, Sahmyook Medical Center, Seoul 02500, Republic of Korea; mapatofu@gmail.com (J.L.); nicehsj1063@naver.com (S.H.); 2Medicine Department Visiting Health Team, Dongdaemun-gu Public Health Center, Seoul 02500, Republic of Korea; 3Department of Physical Medicine and Rehabilitation, Loving Care Clinic, Seongnam-si 13524, Republic of Korea; nnssy0311@ddm.go.kr

**Keywords:** rehabilitation, caregiver burden, robotics, pilot study

## Abstract

**Background:** With South Korea’s growing aging population, the demand for accessible rehabilitation solutions is increasing. Home-based robotic rehabilitation presents a feasible alternative to conventional in-clinic rehabilitation. This study explores the impact of the Rebless robotic rehabilitation device in a home-based setting for people with physical disabilities and their caregivers. **Methods:** We prospectively collected data from individuals with brain disorders or physical disabilities living in Dongdaemun-gu, from August 2023 to March 2024. Participants completed an 8-week rehabilitation program using the Rebless robotic device. Assessments were conducted at baseline and after the eight-week program, measuring motor function, caregiver burden, and quality of life. Exercises were performed three times weekly for at least 90 min total. **Results:** We conducted an intervention with 26 adults with physical or neurological disabilities, of which 20 completed the program. Significant improvements were observed in upper limb function within the elbow exercise group (Fugl–Meyer assessment for upper extremity, *p* = 0.043) and a reduction in caregiver burden across the total groups (Zarit Burden Interview, *p* = 0.003). However, no statistically significant changes were found in balance and mobility measures (Berg balance scale, timed up-and-go, 10 m walk test). **Conclusions:** Home-based robotic rehabilitation demonstrates potential for improving upper limb function and reducing caregiver burden and mental health, proving beneficial to both patients and caregivers.

## 1. Introduction

In 2023, the number of people with disabilities in South Korea reached 2.6 million, representing 5.1% of the total population, the same proportion as in 2022. Among them, 43.7% have physical disabilities, which involve impairments or limitations in mobility, such as amputation, joint disorders, and neurological conditions such as spinal cord injuries; 9.1% have brain lesions, and the proportion of registered individuals with disabilities aged 65 and older has steadily increased, reaching 53.9% [1]. With an aging population and the number of people with disabilities remaining stable, the necessity for appropriate rehabilitation programs to maintain independent living in communities is increasing. While traditional rehabilitation programs have primarily been based in specialized facilities, disabled individuals often face difficulties accessing these services due to their vulnerable health status and lack of accessibility [2,3]. Therefore, most people with disabilities return to their home or community without sufficient rehabilitation time. Home-based rehabilitation programs offer potential solutions by facilitating rehabilitation exercises without requiring visits to medical facilities, reducing both accessibility issues and financial burdens [4,5]. A previous study indicated convenience and being comfortable within the home as two significant advantages of home-based rehabilitation. The disadvantages identified by the subjects included the lack of equipment and floor space and the demotivating effect of the home setting [6]. Another study highlighted the importance of considering technical support and the physical environment at home for home-based rehabilitation [7]. The concept of utilizing robots in home-based rehabilitation has been under discussion since the COVID-19 pandemic era, but there are not many studies on this subject [8,9]. Although various studies have investigated the effectiveness of robot rehabilitation programs within clinical settings [10,11], there is a notable lack of systematic research examining the psychological impacts of home-based robotic rehabilitation programs for individuals with disabilities.

This study aims to evaluate the effectiveness of the Rebless robotic rehabilitation device provided to disabled individuals, focusing on participants’ limb function and quality of life, as well as caregivers’ burden and quality of life before and after the intervention. This study is part of a joint public–private rehabilitation project conducted by Sahmyook Medical Center and Dongdaemun Health Center. Public–private partnerships (PPPs) are recognized globally as effective approaches to addressing healthcare challenges by combining the technological innovation and efficiency of the private sector with the outreach and public health focus of government entities [12]. In this study, the PPP facilitated the delivery of robotic rehabilitation devices to underserved populations, overcoming barriers such as costs and geographical limitations. This model highlights how public–private collaboration can serve as a scalable and sustainable approach to expanding access to cutting-edge rehabilitation technologies. The study aims to evaluate changes in limb function and quality of life using physical examinations and patient surveys, along with additional caregiver surveys conducted before and after the rehabilitation program.

## 2. Materials and Methods

### 2.1. Participants

We prospectively collected data from individuals with brain disorders or physical disabilities living in Dongdaemun-gu who were suffering from muscle weakness, gait disturbances, or upper limb function impairments between August 2023 and March 2024. Participants were recruited through promotional materials created by Dongdaemun Health Center and telephone consultations, targeting individuals with disabilities registered in Dongdaemun-gu.

A total of 26 patients were enrolled in the study, and 6 participants withdrew due to joint pain, failure to meet the exercise volume, or dissatisfaction with the program. Thus, 20 patients completed the intervention. Among the 20 patients, 5 patients performed elbow exercises, and 15 patients performed knee exercises.

This unblinded study included participants selected based on the following criteria: (1) registered as disabled in Dongdaemun-gu; (2) aged 19 or older; (3) able to communication using gesture or verbal language; and (4) able to walk independently and perform daily activities before the onset of their condition.

The exclusion criteria were as follows: (1) inability to communicate due to severe cognitive impairment or aphasia; (2) medical conditions that made it difficult to continue participating in the study; (3) inability to use machines or operate a smartphone; and (4) severe spasticity classified as grade 4 on the Modified Ashworth Scale (MAS). Participants could withdraw from the experiment under the following conditions: (1) the participant or their caregiver wished to discontinue; (2) adverse effects or complications occurred during the use of equipment; (3) a medical issue arose that made continuation of the study difficult; (4) failure to meet the prescribed exercise volume during the program.

### 2.2. Equipment

Rebless is an electric orthopedic exercise device developed by H. Robotics (Figure 1). It is designed to aid patients with joint and muscle paralysis resulting from musculoskeletal and neurological disorders by improving joint range of motion and restoring muscle strength. This versatile device can be used for four joints: the elbow, wrist, knee, and ankle.

The machine offers two modes: the active mode and the passive mode. In the active mode, the motor assists the patient’s movements or applies resistance based on their muscle strength. In the passive mode, the motor passively moves the patient through exercises. In addition to the two modes, settings such as range of motion, repetition count, and resistance and assistance levels can be adjusted up to a maximum level of 10. The device synchronizes with a mobile application, automatically recording exercise data such as range of motion, duration, resistance level, and speed at the end of each session. Usage records can also be remotely monitored using a tablet PC. These records were transmitted to and monitored by staff from Dongdaemun Health Center, who provided participants with feedback on exercise duration and intensity.

Furthermore, the exercise area and intensity for patients were determined by the physician, considering factors such as the paralyzed area, degree of muscle weakness, and participant preferences.

### 2.3. Procedure and Intervention

Participants received the Rebless robotic rehabilitation device after an initial evaluation, which was transported and installed by staff from Dongdaemun Health Center. They performed rehabilitation exercises for at least 30 min per session with Rebless, three times a week (totaling a minimum of 90 min weekly) for eight weeks.

The exercise regions were determined as follows: For patients with hemiparesis or quadriplegia, the intervention was applied to the limb (upper or lower) with the most severe weakness. For patients with paraplegia, the intervention was applied to the side (left or right) with greater weakness. If the muscle strength of all limbs was judged to be nearly equal, the intervention was applied to the region the patient preferred.

The training intensity of Rebless was set as follows: During the initial evaluation, the participants performed training without assistance or resistance. If the participants were able to train with a full range of motion, the resistance level was increased to the maximum level they could tolerate. If training without assistance was not possible, the assistance level was increased until full range of motion training was achievable. Training was conducted at the levels, with the elbow group performing elbow flexion and extension exercises and the knee group performing knee flexion and extension exercises. Subsequently, the participants were monitored remotely via a tablet PC, and if intensity adjustments were needed, feedback was provided to the participants to reset the training intensity.

### 2.4. Clinical Aseessment

All patients underwent the same evaluations at two time points: (1) initial evaluation at the beginning of exercise; and (2) follow-up assessment at 8 weeks after exercise, conducted by same rehabilitation physicians within 1 week of completing the exercise program at either Dongdaemun Health Center or the participants’ homes, depending on their mobility. Initial and follow-up evaluations were conducted at the same time of day, between 2 PM and 5 PM.

Baseline characteristics collected included sex, age, time since injury, disease etiology, neurological level, modified ranking scale (mRS), and MAS. Additionally, treatment duration, degree of range of motion, number of repetitions, level of assistance, and resistance of the device were recorded on the participant’s synchronized mobile app and monitored by the researcher using a tablet PC.

To assess changes in motor function and overall functionality before and after the robotic rehabilitation, the Medical Research Council (MRC) grade was evaluated for all patients. Additionally, the Fugl–Meyer assessment for upper extremity (FMA-UE) [13] was conducted for the elbow exercise group, while the Berg balance scale (BBS) [14], timed up-and-go (TUG) [15], and 10-m walk test (10 mWT) [16] were performed on the knee exercise group. To investigate caregiver burden, the Zarit Burden Interview (ZBI) [17] was used, and health-related quality of life for both patients and caregivers was assessed using the Short-Form 12 Health Survey V2 (SF-12v2) [18]. The Korean versions of ZBI [19] and SF-12v2 [20] were employed in this study, with both measures exhibiting high reliability in previous studies [21,22].

The ZBI is a questionnaire that assesses the level of subjective feelings of burden experienced by caregivers of older persons with dementia and other types of disabilities, with higher scores indicating a greater burden, with scores ranging from 0 to 88. SF-12v2 is an abbreviated version of the 36-Item Short-Form Survey (SF-36), consisting of 12 items across eight categories, and it is divided into two domains: physical component summary (PCS) and mental component summary (MCS). Scores were calculated using the QualityMetric Health Outcomes Scoring Software 4.5, with higher scores indicating better health-related quality of life.

### 2.5. Statistical Analysis

Data analysis was conducted using IBM SPSS ver. 29 for Windows (IBM Corp., Armonk, NY, USA). Statistical significance was defined as a *p*-value of <0.05. Normality was assessed using the Shapiro–Wilk test, which indicated that the data did not follow a normal distribution. Consequently, the Mann–Whitney U test was employed for comparisons between the elbow and knee groups, while the Wilcoxon signed-rank test was applied for comparisons between pre- and post-exercise groups.

## 3. Results

### 3.1. Participants Characteristics

The median age and interquartile range (IQR) of the participants were measured. The median age (IQR) of those who performed the elbow exercise was 45 years (range, 41.0–65.0 years), while for the knee exercise group, it was 56 years (range, 36.0–68.0 years). The median time since injury (IQR) was 8 years (range, 2.5–10.0 years) in the elbow exercise group and 15 years (range, 4.0–26.0 years) in the knee exercise group. In the elbow exercise group, there was one patient with ischemic stroke, three patients with hemorrhagic stroke, and one patient with spinal cord injury. In the knee exercise group, there were three patients with ischemic stroke and six with hemorrhagic stroke. Additionally, six patients in the knee exercise group had other conditions, including cerebral palsy, avascular necrosis, Kennedy’s disease, and hereditary spastic paraplegia. Demographic data are presented in Table 1.

### 3.2. Clinical Assessment of Subjects

Table 2 presents the baseline and post-treatment results. The improvement in the FMA-UE scores for the elbow group (*p* = 0.043) indicates that robot-assisted rehabilitation can be particularly beneficial for upper limb function in patients with chronic disabilities. Moreover, the significant reduction in ZBI scores for the knee group (*p* = 0.002) and total group (*p* = 0.003) and the improvement trend observed in the elbow group, though not statistically significant, indicate that the home-based robot rehabilitation program positively impacts caregiver burden. This significant finding underscores the potential for home-based rehabilitation to ease the burden on caregivers. Additionally, the MCS score of caregivers exhibited a statistically significant improvement (*p* = 0.046), suggesting a positive psychological impact on caregivers.

BBS, TUG, and 10 mWT showed trends toward improvement in the knee exercise groups; however, no statistically significant improvements were observed.

## 4. Discussion

The primary findings of this study are the significant improvement in upper limb function, as assessed by the FMA-UE results, and the reduction in caregiver burden, as indicated by the ZBI scores. Additionally, the significant improvement in the MCS scores highlights the psychological benefits for caregivers, suggesting that reduced physical burden may also contribute to better mental health outcomes among caregivers. These results support the potential efficacy of home-based robotic rehabilitation in enhancing both patient outcomes and caregiver well-being.

The improvement in FMA-UE scores observed in the elbow group suggests that robot-assisted rehabilitation can be particularly beneficial for upper limb function in patients with disabilities. This aligns with previous studies that have demonstrated significant improvements in FMA-UE scores in hospital-based robotic rehabilitation [23,24,25]. Therefore, this study confirms that similar outcomes can be achieved in a home-based setting, emphasizing the convenience and accessibility of home-based rehabilitation programs.

The reduction in ZBI scores suggests that the home-based robot rehabilitation programs can alleviate caregiver burden. Home-based programs allow caregivers to engage actively in the rehabilitation process, potentially fostering a sense of empowerment and control, which may contribute to reduced stress and better outcomes for both patients and caregivers. Supporting this, Chen et al. [26] demonstrated that both home-based and hospital-based rehabilitation programs were effective in enhancing the quality of life among participants, suggesting that home-based approaches can offer comparable benefits to traditional, facility-based care in terms of life quality improvements. Additionally, Wang et al. [27] found that long-term home-based rehabilitation programs for patients with acute myocardial infarction were effective in improving quality of life and reducing anxiety levels, further underlining the psychological benefits of home-based rehabilitation.

Analysis of the 22 items in the ZBI revealed significant improvements in specific questions (*p* < 0.05), including questions on privacy, life control, uncertainty, and the overall burden of caring for the relative. These findings, combined with the MCS of caregiver improvements, suggest that a home-based rehabilitation program can contribute positively to the mental well-being of caregivers by reducing both the physical and emotional strains associated with caregiving.

This study also underpins important social implications. Many individuals with disabilities face challenges in accessing consistent rehabilitation due to mobility limitations, financial burdens, or lack of family support [2,28]. These barriers can lead to social isolation, a known risk factor for mental health problems [29]. Home-based rehabilitation programs, supported by public initiatives, could address these challenges by providing regular and personalized rehabilitation services in the home environment. Such programs not only improve physical health but also foster social engagement and reduce feelings of isolation [30]. Therefore, government-funded initiatives offering robotic rehabilitation devices and regular healthcare provider support could be particularly beneficial for individuals living alone, helping to mitigate social isolation and its associated consequences.

The importance of accessible and home-based rehabilitation solutions has been underscored during global crises such as the COVID-19 pandemic [31]. Telerehabilitation strategies, including remote assessments and therapeutic interventions, played a critical role in maintaining continuity of care when in-person services were severely restricted [32]. By leveraging such strategies, home-based robotic rehabilitation programs can ensure consistent care while mitigating the challenges posed by limited access to traditional rehabilitation facilities.

Despite these positive results, the lack of significant improvements in balance and mobility, as measured by BBS, TUG, and 10 mWT, may be attributed to the nature of the intervention, which focused on single joint rehabilitation rather than whole-body functional training. These tests primarily evaluate overall functional performance, such as balance and walking, which involve multiple joints and complex neuromuscular coordination. As a result, they may not fully capture the localized benefits provided by the intervention. Additionally, the relatively short 8-week intervention period may have been insufficient to observe significant changes in lower limb functions such as balance and walking, as these typically require extended rehabilitation. The chronic nature of conditions in the knee group, with a median time since injury of 10 years, may have contributed to the slower response to rehabilitation interventions, as chronic cases often demonstrate reduced plasticity and slower recovery rates.

Furthermore, the diverse pathologies among the participants may have led to differing improvements, further complicating the interpretation of results. This heterogeneity was partly due to the study being conducted in a community setting rather than a hospital-based setting, which limited recruiting a more homogeneous patient population. While community-based studies reflect real-world conditions and include participants with varying levels of disability, this diversity introduces variability that makes it challenging to standardize assessments and interpret outcomes uniformly.

The small sample size (20 participants) could limit the statistical power. Future studies with larger samples and longer follow-up periods are recommended to better capture the potential long-term benefits of robotic rehabilitation on mobility. Also, the variability in the participants’ adherence to the home-based regimen, differences in motivation, and ability to follow the program could have impacted the results. Future research should consider enhanced adherence monitoring, potentially through real-time monitoring of supervisors or frequent check-ins, to ensure consistent program implementation.

An additional limitation is that while the robotic devices were provided to the participants free of charge through a public–private partnership facilitated by government agencies in this study, it may be financially challenging for participants to bear the cost of the robotic devices if hospitals were to take charge of the treatment. Therefore, public support for people with disabilities in society will be essential for home-based robotic rehabilitation.

Despite these limitations, this study contributes valuable insights into the potential benefits of home-based robotic rehabilitation programs. The results suggest that these programs are not only feasible but can also yield meaningful improvements in patient outcomes, particularly for upper limb function, while simultaneously reducing caregiver burden.

## 5. Conclusions

To the best of our knowledge, this is the first study in South Korea to evaluate the effectiveness of a home-based robotic rehabilitation program for individuals with disabilities conducted without any intervention from a supervisor during the exercise program. The observed improvements in upper limb function and caregiver burden reduction may be associated with the home-based robotic rehabilitation program. Home-based rehabilitation using robotic devices has shown potential as a complementary approach to conventional in-clinic rehabilitation programs, indicating the need for further research to validate and expand upon these findings. Future research is needed to confirm these findings and explore the underlying mechanisms. Research should also aim to optimize the balance between patient autonomy and consistent rehabilitation efforts for maximizing program effectiveness.

## Figures and Tables

**Figure 1 healthcare-13-00078-f001:**
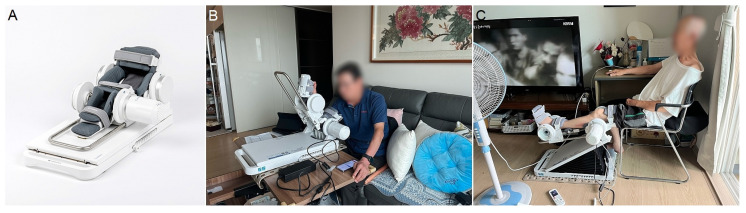
(**A**) A robotic rehabilitation device, Rebless. (**B**) Patient exercising elbow with the robotic device at home. (**C**) Patient exercising knee with the robotic device at home.

**Table 1 healthcare-13-00078-t001:** Demographic data of patients.

Characteristic	Elbow	Knee	Total	*p*-Value
No. of patients	5	15	20	
Sex				0.672
Male	4	10	14	
Female	1	5	6	
Age (yr)	45 (41–65)	56 (36–68)	55 (40–68)	0.866
Time since injury (yr)	8 (2.5–10.0)	15 (4.0–26.0)	10 (3.25–24.0)	0.054
Injury type				
Ischemic stroke	1	3	4	
Hemorrhagic stroke	3	6	9	
Cerebral palsy	0	3	3	
Spinal cord injury	1	0	1	
Avascular necrosis of femoral head	0	1	1	
Kennedy’s disease	0	1	1	
Hereditary spastic paraplegia	0	1	1	
Neurological level				
Hemiparesis	3	5	8	
Paraplegia	0	1	1	
Quadriparesis	2	9	11	
Modified Rankin Scale	4 (3–4)	4 (2–4)	3 (2–4)	0.142
Modified Ashworth Scale				0.195
0	1	6	7	
1	2	8	10	
2	2	1	3	

Values are presented as median (interquartile range). *p*-value represents statistical comparisons between elbow and knee groups.

**Table 2 healthcare-13-00078-t002:** Clinical outcome in each group pre- and post-program.

		Elbow			Knee			Total	
	Pre	Post	*p*-Value	Pre	Post	*p*-Value	Pre	Post	*p*-Value
MRC	3 (2.25–3.75)	3.5 (2.75–3.50)	0.317	3 (2.5–3.5)	3.5 (2.5–3.5)	0.317	3 (2.5–3.5)	3.5 (2.5–3.5)	0.166
FMA-UE	37 (23.5–51.0)	45 (28.0–58.5)	0.043 *						
BBS				12 (4–44)	17 (4–38)	0.192			
TUG				20.46 (14.00–46.50)	20.17 (13.33–41.83)	0.093			
10 mWT				18.83 (14.27–37.64)	20.54 (12.12–31.20)	0.575			
ZBI	56 (31.75–63.00)	44.50 (28.00–60.25)	0.581	29 (24.25–46.00)	22.5 (15.00–26.25)	0.002 *	33 (24.75–48.50)	24 (18.50–35.25)	0.003 *
PCS	29.66 (28.31–33.75)	35.93 (22.58–41.32)	0.686	33.62 (24.55–38.96)	31.65 (26.14–35.13)	0.507	31.73 (26.65–38.03)	31.65 (24.67–38.72)	0.845
MCS	54.73 (35.27–57.69)	48.74 (31.47–59.77)	0.686	57.90 (40.59–64.02)	60.29 (50.09–66.66)	0.249	55.03 (40.45–63.30)	54.65 (44.70–65.68)	0.472
PCS_c	55.18 (35.94–56.24)	44.46 (41.72–50.62)	0.465	53.07 (45.33–55.00)	49.75 (35.12–53.23)	0.311	53.31 (45.33–55.51)	49.27 (40.54–52.63)	0.193
MCS_c	44.20 (32.16–49.35)	38.51 (31.82–53.00)	1.000	52.47(47.03–58.15)	60.59 (53.84–62.52)	0.046 *	50.95 (44.20–56.76)	58.10 (41.81–61.24)	0.093

Values are presented as median (interquartile range). MRC, Medical Research Council; FMA-UE, Fugl–Meyer assessment of upper extremity; BBS, Berg balance scale; TUG, timed up-and-go; 10 mWT, 10 m walking test; ZBI, Zarit Burden Interview; PCS, physical component summary; MCS, mental component summary; PCS_c, PCS of caregiver; MCS_c, MCS of caregiver, * Significant difference among groups (*p* < 0.05).

## Data Availability

The data presented in this study are available on request from the corresponding author. The data are not publicly available due to privacy restrictions.

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
