# Peer review of "The Effect of Home-Based Robotic Rehabilitation on Individuals with Disabilities in Community Settings: A Pilot Study"

_healthcare, 2025, doi:10.3390/healthcare13010078_

Round 1

Reviewer 1 Report

Comments and Suggestions for Authors

The presented article aims to analyze the impact of home-based robotic rehabilitation on individuals with disabilities in community settings.

Structurally, the article meets all the requirements of an experimental study of this type. However, there are some fundamental aspects that distort the purpose of the document:

1.      In my opinion, it is not truly robotic rehabilitation; rather, it is simply the use of an electronic device that improves joint range of motion. These devices are designed to provide controlled and progressive joint mobilization, which helps restore flexibility, strength, and functionality to the affected joint, specifically in musculoskeletal (MSK) disorders. A robotic device actively interacts with and responds to its environment. Moreover, it may include AI or machine learning for decision-making and adaptation. In contrast, an electronic device has minimal interaction with its environment.

2.      The lack of significant differences in balance and mobility, as measured by BBS, TUG, and 10mWT in the patients, is likely because these tests evaluate overall function rather than the isolated movement of a single joint. It would be overly reductive to consider such a possibility, and if it were true, these devices alone would be sufficient to reduce disability. Undoubtedly, improving the range of motion of intermediate joints of the limbs favors functional independence. This improvement could provide significant assistance to caregivers of these patients, which may explain the significant differences observed in the ZBI test for the lower limbs.

3.      There are too many variables being evaluated among a small number of subjects with varied pathologies. Although the evaluated parameters are related to disability, they do not necessarily allow for its proper assessment. Specific tests exist to evaluate disability; however, having few patients with different conditions makes it challenging to identify a test that can evaluate them homogeneously.

4.      There is a lack of information about the conditions under which the level of assistance and resistance of the device was modified. What was the criterion for reprogramming the treatment session? Finally, there is no bibliographic support for the use of the equipment, nor is there sufficient scientific evidence to support the discussion.

I believe the study should be presented from another perspective. For example, it would be interesting to analyze the impact of single-joint range of motion on the caregiver's workload in patients with disabilities. Considering the context provided, the authors likely have a different vision for the focus of their study. However, it is crucial to carefully select appropriate tests and determine an adequate number of patients (or patient groups), ideally with homogeneous characteristics.

Reviewer 2 Report

Comments and Suggestions for Authors

This pilot study presents promising evidence on the feasibility of home-based robotic rehabilitation for improving upper limb function and reducing caregiver burden. Here are my comments for improvement: 

1. The authors mention the collaboration between public and private institutions but do not discuss the significance or novelty of this partnership. Please elaborate on the importance of public-private partnerships in enhancing accessibility to rehabilitation technologies.

2. I suggest the authors discuss the feasibility of incorporating this device into telerehabilitation strategies. For example, can the data collected by the device’s synchronized app be shared with rehabilitation specialists remotely for real-time monitoring, feedback, and guidance? Exploring this capability would significantly improve the scalability and utility of the intervention.

3. One of the critical lessons from the COVID-19 pandemic was the necessity of accessible and home-based rehabilitation solutions when in-person care is unavailable. The manuscript does not discuss this aspect, which is highly relevant to the utility of robotic rehabilitation devices. I encourage the authors to elaborate on how this device could serve as an essential tool during extreme conditions, such as a global pandemic. Highlighting this aspect would underscore the device’s potential to improve healthcare delivery resilience during crises.

Please see this paper: Mihai EE, Popescu MN, Beiu C, Gheorghe L, Berteanu M. Tele-Rehabilitation Strategies for a Patient With Post-stroke Spasticity: A Powerful Tool Amid the COVID-19 Pandemic. Cureus. 2021 Nov 2;13(11):e19201. doi: 10.7759/cureus.19201. PMID: 34877194; PMCID: PMC8642141.

Reviewer 3 Report

Comments and Suggestions for Authors

Thank you for allowing me to review the study titled: “The effect of home-based robotic rehabilitation on individuals with disabilities in community settings: a pilot study”. I aimed to give objective criticism, hoping to achieve that goal.

This pilot study examines the impact of the Rebless robotic rehabilitation device in a home-based setting on motor function, caregiver burden, and quality of life of individuals with physical disabilities. This paper is well-written but has significant limitations, including a small participant pool and questions about the study's originality. Additionally, the introduction, methods, and discussion sections require substantial revisions. 

Specific comments can be found in the file.

Round 2

Reviewer 1 Report

Comments and Suggestions for Authors

After reading the second version, I have confirmed that the paper has been significantly improved; therefore, it is suitable for publication.

I agree with the authors in highlighting the importance of working on intermediate joints to improve functionality in patients with disabilities. Particularly, how this improvement can reduce dependence on caregivers. Although the effect is clinically and statistically significant (especially in the lower limbs), it is important to note that it also has its limitations concerning disability. In future studies, it would be interesting to explore the level of adherence achieved once the maximum functional level is reached. Perhaps by incorporating other devices (such as virtual reality or serious games), the benefits achieved could be sustained for a longer period.

Reviewer 3 Report

Comments and Suggestions for Authors

The manuscript titled "The effect of home-based robotic rehabilitation on individuals with disabilities in community settings: a pilot study"   has been sufficiently improved to warrant publication in Healthcare.